# Fabrication and Characterization of a Multifunctional Coating to Promote the Osteogenic Properties of Orthopedic Implants

**DOI:** 10.3390/ma16196608

**Published:** 2023-10-09

**Authors:** Serap (Gungor) Koc, Tuba Baygar, Selma Özarslan, Nurdan Sarac, Aysel Ugur

**Affiliations:** 1Department of Mechanical Engineering, Faculty of Engineering, Van Yuzuncu Yil University, 65080 Van, Turkey; 2Research Laboratories Center, Mugla Sitki Kocman University, 48000 Mugla, Turkey; tubaygar@mu.edu.tr; 3Department of Physics, Faculty of Science, Hatay Mustafa Kemal University, 31060 Hatay, Turkey; selmaozarsln@gmail.com; 4Department of Biology, Faculty of Science, Mugla Sitki Kocman University, 48000 Mugla, Turkey; nsarac@mu.edu.tr; 5Section of Medical Microbiology, Department of Basic Sciences, Faculty of Dentistry, Gazi University, 06500 Ankara, Turkey; ayselugur@gazi.edu.tr

**Keywords:** hydroxyapatite, bioglass, chitosan, heparin, Ti6Al4V alloy

## Abstract

Titanium-based alloys are used in orthopedic applications as fixation elements, hard tissue replacements in artificial bones, and dental implants. Despite their wide range of applications, metallic implant defects and failures arise due to inadequate mechanical bonding, postoperative clotting problems, aseptic loosening, and infections. To improve the surface bioactivity and reduce the corrosion rate of the Ti6Al4V alloy, multi-layered coatings (HAp, BG, Cs, and Hep) were applied via electrophoretic deposition (EPD). XRD images showed the presence of HAp within the coating. In vitro investigation: cell line NIH-3T3 fibroblasts were seeded on the non-coated and coated Ti6Al4V substrates, and their cellular behavior was evaluated. The results indicated that the HApBGCsHep coating could enhance the adhesion and proliferation of NIH 3T3 cells. In addition, the potentiodynamic polarization results are compatible with the in vitro outcome.

## 1. Introduction

Human scaffolds made up of bones, joints, and cartilage, which are responsible for providing structural support for muscles and tendons that allow movement [1]. Long bones are hard, compact bones that provide structure, strength, and mobility to the upper and lower extremities. They have a wide internal cavity covered by the inner bone surface, the endosteum, and filled with bone marrow [2]. Researchers have reported structures like transverse cortical vessels, which also consist of human limb bones. Transverse cortical vessels are formed of arterial or venous markers and transport neutrophils. These carry oxygenated fresh blood via arteries into the bone and transport used blood out via veins [3].

The activity of any organ is dependent on effective blood circulation. Unfortunately, thrombosis is one of the most important health problems concerning blood-contacting metallic implants. Mostly, after implantation on the implant surface, adhered platelets may be activated due to the adsorption of plasma proteins, calcium, and platelet adhesion, leading to the coagulation cascade and then thrombosis [4]. In this respect, implant surfaces coated with organic/inorganic hybrid coatings with heparin may be an effective approach to preventing thrombosis. 

Because of its anticoagulant properties, the negatively charged linear polysaccharide heparin is frequently used in drug-eluting stents. As a result, various heparin-coated systems have been explored. One example is a drug-eluting stent dip coated with hydrophobic heparinized polyacrylic acid-N-hydrosuccinimide. In particular, it has been reported that the heparin-coated layer decreased platelet adhesion and improved hemocompatibility. Moreover, in vivo tests revealed that the coated stents prevent restenosis and smooth muscle cell proliferation [5]. 

In biomedical applications, the polycationic natural polymer chitosan, consisting of β-glucosamine and N-acetylglucosamine, is frequently used due to its biocompatible, antibacterial, bioactive, and biodegradable properties [6]. The important features of chitosan are its sensitivity to enzymatic degradation, improved cellular adhesion, and decreasing fibrous encapsulation [7,8]. Mostly, chitosan is combined with other antimicrobial ions to increase its antibacterial properties and immobilize proteins, nucleic acids, and virus particles. There are several studies that have revealed that chitosan–heparin nanocarriers are appropriate for their long-term anticoagulant activity and ability to bind antithrombin from plasma [9,10].

Recently, a need has emerged for improved clinical orthopedic implants designed to solve bone problems associated with aging. Zinc-based biodegradable implants have the potential to serve as orthopedic implants for the treatment of complex bone disorders due to their favorable mechanical and degradation specifications. However, numerous post-processing techniques are required to enhance the microstructure of zinc-based biodegradable implants [11]. Among metallic implants, titanium and its alloys are frequently used in orthopedic applications due to their prominent mechanical properties and excellent biocompatibility [12]. A titanium dioxide passive layer formed on the surface of titanium alloys can improve the biocompatibility and corrosion resistance of the implants [13]. However, due to poor osteoinductive properties, fibrotic encapsulation generated around the implants and the toxic effect of vanadium release into the host tissues will affect the long-term stability of the prosthesis [14]. Despite their wide range of applications, metallic implant defects and failures arise due to inadequate mechanical bonding, postoperative clotting problems, aseptic loosening, and infections [15]. The main reason for aseptic loosening is implant micromotion owing to gaps in the implant–tissue interface. Therefore, the application of ceramic/biopolymer surface coatings is intended to increase the osteointegration properties of metallic implants. In a total hip replacement, the broken bone is removed and replaced with implants. The success of the operation depends on the tight attachment of the implant to the damaged bone and the prevention of corrosion and infections. One of the methods to ensure this is to make the implant surface rough, or to coat the implant with special substances that will allow better adhesion to the implant. Hence, implants can be attached directly to the bone without using any filling material. 

Ceramic bioglass coatings can provide increased mechanical properties, high corrosion resistance, and enhanced bioactivity [16]. Bioactive glasses, known as osteoproductive materials, promote the development of new tissues by releasing ions. Bioglasses produce specific biological responses at in vivo interfaces between soft tissue and bone. In particular, bioactive glasses containing CaO-SiO2-P_2_O_5_ can be attached to soft and hard tissues without intermediate fibrous layers. In vivo implantations have shown that these compositions do not produce local or systemic toxicity, inflammation or body responses [17]. However, the application of bioglass is limited in implant coatings due to their lower tensile strength and fatigue resistance [18]. Thermal expansion coefficient parameters differ between the interface of bioglass coatings and titanium substrates, which leads to cracking during the sintering process. Recently, bioglass coatings have been loaded with different ceramic particles to improve microstructural properties. Hydroxyapatite (HAp), similar to the inorganic component of bone, is widely used in orthopedic and dental applications due to its osteoconductive activities. Therefore, doping HAp with bioglass is an appropriate method to improve the cell attachment, adhesion, mechanical, and antibacterial properties of the coatings [19].

In recent years, electrophoretic deposition (EPD) has been used to produce uniform coatings at low temperatures. The significance of this technique comes from its simplicity, low equipment cost, easy control of thickness, and the possibility of it being applied to complex-shaped implants and porous structures [20]. In the EPD method, surface-charged powder particles are deposited on an electrode under an electrical field [21]. Considering the significance of HAp, bioglass (BG), chitosan (Cs), and heparin (Hep), the purpose of this study was to develop novel multifunctional composite HApBGCs coatings loaded with heparin on titanium substrates (CpTi) to improve surface properties, corrosion resistance, clotting problems, biocompatibility, and antibacterial activity. To investigate the effect of the binary combinations of HAp, BG, and Cs loaded with heparin dopants, they were evaluated via XRD, SEM-EDX, FTIR, and biocompatibility tests. 

## 2. Materials and Methods

### 2.1. Materials

In order to produce the doped HA coating, the following chemicals were used: Ca(NO_3_)_2_.4H_2_O (Isolab, Eschau, Germany, ≥99%), SiC_8_H_20_O_4_ (Merck, Rahway, NJ, USA, ≥99), HNO_3_ (Sigma-Aldrich, Saint Louis, MO, USA, ≥65%), C_6_H_15_O_4_P (Alfa Aesar, Haverhill, MA, USA, ≥98%), C_2_H_6_O (Sigma-Aldrich, ≥99.8%), (C_6_H_11_NO_4_)_n_ chitosan (TCI), heparin (Kocak Farma, Istanbul, Turkey), and double-distilled water.

### 2.2. Bioglass, Chitosan and Heparin-Doped Hydroxyapatite (HAp, BG, Cs, Hep) Suspension

The co-precipitation process was used to produce the BG-doped HAp powder with the purpose of creating a homogenous coating [22,23]. The BG precursor suspension prepared as SiC_8_H_20_O_4_ was dissolved in 0.1 M nitric acid and stirred for 30 min. C_6_H_15_O_4_P and Ca(NO)_2_.4H_2_O were added in the order specified and stirred at 70 °C for 24 h. The resulting solution was filtered and sintered at 600 °C for 2 h. The suspension for EPD of HApBG was prepared using ethanol (C_2_H_5_OH), distilled water, and acetic acid (CH_3_COOH). Before the EPD process, 0.5 g/L chitosan was dissolved in acetic acid (0.2 vol%) in a dilute solution [24]. The solution described before was also mixed with 4000 IU/mL oksapar (enoksaparin, low-molecular-weight heparin) (Kocak Farma). To prevent hydrogen bubbles and obtain uniform coatings, ethanol, chitosan, acetic acid, and distilled water in particular were used to prepare the EPD solution [25].

### 2.3. Preparation of Coatings

In this study, Ti6Al4V plates (Grade 5 ELI) were used as a substrate, with a cross-section of 20 × 20 mm^2^. The thickness of the substrate was 0.2 cm. Substrates were wet-abraded by using SiC papers (400–1200 grits) and then ultrasonically cleaned for 15 min with ethanol and distilled water. We applied the electrophoretic deposition process to the Ti6Al4V substrates with a Class DC Power Supply 305D to obtain double-uniform coatings. EPD coating was applied to suspensions prepared by HAp, HApBG (Figure 1), and HApBGCsHep (CsHep dip-coated). Afterwards, we used two Ti6Al4V disks as electrodes. The distance between two electrodes was set at 20 mm. We applied EPD at room temperature (30 V for 1 min) and repeated it twice. In order to enhance the interaction between Cs-Hep and Ti6Al4V interface, BGHAp was initially coated on the Ti6Al4V alloy via an EPD method, and then air-dried. Finally, HApBG-coated substrates were dip-coated in the Cs-Hep solution and air-dried. 

### 2.4. Characterization Methods

The crystal structure of the synthesized coatings was obtained using XRD (PANalytical Empyrean, PANalytical B.V., Almelo, The Netherlands). The surface morphology of the HAp, BG, Cs, and Hep coatings was assessed via scanning electron microscopy SEM (Zeiss Gemini Sigma 300, Zeiss, Oberkochen, Germany). The distribution of the elemental constituents was investigated via EDX (Zeiss, Gemini, Sigma 300). The HAp, BG, Cs, and Hep coatings were analyzed in the 400–4000 cm^−1^ spectral range via FTIR using a Thermo Scientific, Waltham, MA, USA/S10FT-IR spectrometer.

### 2.5. Cell Culture and Morphology

Biocompatibility was assessed by growing NIH-3T3 fibroblasts (American Type Culture Collection, ATCC, Manassas, VA, USA) on the discs, and their viability was examined using a 3-[4,5-dimethylthiazol-2-yl]-2,5-diphenyltetrazolim bromide (MTT) colorimetric assay [26]. The NIH 3T3 mouse embryonic fibroblast cell line provided by the American Type Culture Collection (ATCC, Manassas, VA, USA) was grown in Dulbecco’s Modified Eagle’s Medium (DMEM)-high glucose, supplemented with 10% heat-inactivated fetal calf serum (FCS), L-glutamine (2 mM), and 1% antibiotic-antimycotic solution (10.000-unit penicillin, 10 mg streptomycin, and 25 μg amphotericin B per mL). Cells were maintained at 37 °C in a humidified atmosphere of 5% CO_2_. NIH 3T3 cells were amplified in a 25 cm^2^ flask at semi-confluence and detached using trypsin-EDTA (Gibco, ThermoFisher Scientific, Paisley, UK). Discs (S_1_: control group, uncoated Ti6Al4V, S_2_: HAp+Bioglass, S_3_: Chitosan+Heparin, S_4_: HAp+Bioglass+Chitosan+Heparin) were placed on the bottom of a 6-well plate. The cells grown on the well bottom (without disc) are considered to represent 100% viability (blank) and the uncoated titanium disc was used as a negative control [27]. Some 1 × 10^4^ cells/mL were seeded on each disc surface for test samples and the negative control, as well as on the well bottom for the control well. Briefly, 200 µL of MTT (5 mg/mL, prepared in phosphate-buffered saline) was added to each well after 24 h of incubation and again incubated at 37 °C for an additional 3 h. The medium containing MTT was then poured off, and 200 µL of DMSO was used to solubilize the formed formazan crystals in each test sample surface and negative control well. Plates were put in an orbital shaker for 15 min, and the absorbances were measured at 540 nm using a microplate reader (Thermo Scientific Multiskan FC, Thermo Fischer, Vantaa, Finland).

### 2.6. Potentiodynamic Polarization

The in vitro corrosion behavior of the coated Ti6Al4V alloys was obtained in the Hank solution (balanced salt solution, BSS), and measurements were taken in this solution, referred to as HBSS (Hank’s balanced salt solution). The HBSS supplemented with glucose used in this research is detailed in Table 1 (pH 7.4).

All in vitro corrosion measurements of coatings were conducted at 37 ± 1 °C by a potentiostat/galvanostat CH Instruments (602E) using the potentiodynamic polarization (PDS) technique. In vitro corrosion tests were performed using a three-electrode corrosion cell containing the coated Ti6Al4V substrates as the working electrode, a platinum (Pt) electrode as the counter electrode, and an Ag/AgCl electrode as a reference electrode. All measurements were taken at least three times. The potential range to be scanned was set at −200 mV to 200 mV of corrosion potential, and the scanning rate was 1 mVs^−1^. The tafel polarization analysis was carried out at a scan rate of 1 mV/s from 350 mV to 1000 mV around the open circuit potential (OCP). The current density and corrosion potential parameters of the samples were determined using the tafel extrapolarization method. According to Baboian corrosion rates of samples were calculated with the following equation [28].
$$CR=\frac{K\times i_{corr}\times EW}{D}$$
where *K* is 3.7.10^−3^, *i_corr_* is the corrosion current density (µA/cm^2^), *EW* is the equivalent weight, *D* is the density (g/cm^3^), *CR* is the corrosion rate in mm/yr, *K* = 0.00327 is the unit conversion constant, and ρ is the density of samples in g/cm^3^.

## 3. Results and Discussion

### 3.1. Coating (Physical) Characterization

The crystallinity of the HAp, BG, Cs, and Hep coatings was evaluated using the XRD method, and the pattern is given in Figure 2. The XRD patterns of HApBGCsHep coatings are highly compatible with the main diffraction peaks of the hexagonal crystalline phase HAp, JCPDS # 09-0432. It is obvious from Figure 2 that the most intense peaks are HA (211), (002), and (112). The major peaks indicating the formation of the apatite structure of HA and the peak at (211) are wide enough to cover the intense peaks at (112) and (300).

The peaks on all coatings are broader. The coating’s nanoscale structure is the primary cause of these broader peaks. The HAp crystallinity phase has been shown to be suitable for implant applications [29]. The lack of BG and Cs-related peaks in the XRD data, which is likely due to the BG precursors’ trace amounts, showed that Cs had accumulated in semi-crystalline form. Moreover, the CS peaks also disappeared as a result of the effect of inorganic phases on the CS bonds between -OH and -NH_2_ hydrogen molecules [30].

The SEM and SEM-EDX spectra of the powder HApBG, CsHep, and HApBGCsHep coatings are given in Figure 3. Si nanoparticles are not obvious on the HApBGCsHep coating, most probably due to the scale of the SEM images. The SEM-EDX data, however, proved that Si was present in the coatings. Similarly, in all coatings, no cracks or spherical agglomerates were obtained [31]. 

Figure 4c demonstrates the FTIR spectra of the functional groups present in the HApBGCsHep coating. As can be seen from Figure 4, there are different vibrational modes similar to phosphates and hydroxyl groups. Hydroxyl stretch (OH^−^) is observed at 3572 cm^−1^ in the spectra of the HApBGCsHep coating, while the sharp peaks at 565, 603, and 633 are assigned to v_4_ of PO_4_^3−^. v_3_ of PO_4_^3−^ is responsible for the peaks at 1025 and 1091. The carbonate band peaks of 1331, 1455, and 1647 belong to the v_3_ vibration mode of the surface carbonate ions, as in cp-HA. The phosphate v_1_ band is observed at 958 cm^−1^ [32]. These results support the XRD and verify the presence of HA in the coating powder. 

### 3.2. In Vitro Cell Behaviour (Scanning Electron Microscopy (Biocompatibility))

Cell adhesion onto the discs and the cell morphology were investigated using scanning electron microscopy (SEM). NIH 3T3 cells were maintained as described above, and cells were seeded onto the discs (S_1_: control group, uncoated Ti6Al4V, S_2_: HA+Bioglass, S_3_: Chitosan+Heparin, S_4_: HA+Bioglass+Chitosan+Heparin) in 6-well culture plates at a density of 1 × 10^4^ cells/well for direct cell adhesion observation. After 24 h of incubation, the culture media were removed, and the specimens were fixed for 1 h at room temperature with a 2.5% glutaraldehyde solution. After fixation, the discs were rinsed with phosphate-buffered solution (PBS, pH 7.4) and dehydrated in increasing concentrations of ethanol (50, 70, 90, and 96%; 10 min for each concentration). Specimens were air-dried and coated with gold (Emmitech K550, Ashford, United Kingdom before being examined with SEM (JEOL JSM-7600F; JEOL Ltd., Tokyo, Japan). Figure 5 shows the viability of the cells grown on S_2_ (HAp+Bioglass), S_3_ (Chitosan+Heparin), S_4_ (HAp+Bioglass+Chitosan+Heparin) coatings, and S_1_ (uncoated) Ti disc. Cells grown on polystyrene plate were considered to be 100%. The cells seeded on the Ti-discs (negative control) displayed 84.44 ± 2.31% viability.

The viability of cells seeded on S_4_-coated discs was higher when compared to that of the cells in contact with S_1_, S_2_, and S_3_. The lowest viability amount was observed for CsHep-coated discs (S_3_). HApBGCsHep-coated discs (S_4_) proved to have more biocompatibility than those with only a HApBG coating (S_2_) and only a CsHep coating (S_3_). The cell proliferation results indicated that HApBG (S_2_)-coated and HApBGCsHep (S_4_)-coated surfaces have good cytocompatibility, with proliferation rates over 85% of the control. According to ISO 10933-5:2009, if the viability rate is reduced to <70% of the blank, it has cytotoxic potential [33]. Within the present study, none of the cell viability values of the discs were lower than 75%. Instead, the HApBGCsHep coating showed an increase in cell viability when compared with the blank group, and its cell viability rate was significantly higher than that of the Ti-6Al-4V disc [34]; it has been reported that such an increase indicates that the coating process resulted in superior stimulation of cell proliferation compared with bare Ti-6Al-4V discs.

The most important objective of an implant material is to improve biocompatibility. The coating of a bioactive material improves biocompatibility and prevents ion release from the metallic substrate, which results in reduced mechanical failure [35]. To evaluate the cellular behavior of the coated discs, fibroblast cells were seeded on bare Ti6Al4V and the coated Ti6Al4V substrates. 

Cell adhesion is an entire progress of cell attachment, filopodial growth, cytoplasmic webbing, the flattening of the cell mass, and the ruffling of peripheral cytoplasm [36]. 

Chitosan is a natural biocompatible cationic polysaccharide, and heparin is a glycosaminoglycan polyanion that has strong hydrophilicity [37]. The stability of polymer layers on the surface of solid substrates is a major concern in the surface modification of solid substrates [38].

In the present study, cell attachment onto the surface of Cs-Hep-coated discs was lower than in the other groups. Similarly, Follman et al. [39] reported a decrease in fibroblast cells in their study, in which heparin and different chitosan derivatives were used as coating materials to inhibit the natural inflammatory response to implants. They found that heparin and chitosan derivatives were biocompatible, and at the same time, they strongly hindered the proliferation speed of fibroblasts due to the ionic strength of the polyelectrolyte solutions [39].

In another study, Chupa et al. [40] indicated that while chitosan alone supported cell attachment and growth, glycosaminoglycans (GAG)–chitosan materials inhibited the spreading and proliferation of vascular endothelial and smooth muscle cells in vitro. They suggested that the limited cell spreading seen on the GAG–chitosan surfaces was a result of poor adhesion, which was caused by GAG release from the GAG–chitosan complex [40].

Most recently, many new types of biodegradable composites have been developed for orthopedic applications. Ceramic biomaterials such as calcium sulfate, hydroxyapatite (HA), and bioactive glass (BG) have been used for making biodegradable polymer composites [40]. Good biocompatibility is a required feature of biomaterials, as they have potential to be applied as implants into living tissue [34]. The cell morphology, growth behavior, and adhesion of the coated discs and bare Ti6Al4V discs were investigated using SEM (Figure 6). NIH 3T3 cells spread over the surface of the coated and non-coated Ti6Al4V discs. When compared to Ti6Al4V discs, cells seeded onto the coated discs were found form network-like structures. Cells were in close contact with each other, having many dorsal ruffles. Cells seeded onto the coated discs had polygonal or rounded shapes, while the cells on the Ti6Al4V discs were bipolar-spindle-like shaped. At higher magnifications, SEM observation found many filopodia extensions, which indicated the good proliferation of the fibroblasts. Those extensions enhance the adhesion of the cells to the surface of the material and improve cell migration [34]. The HApBGCsHep-coated discs exhibited more extensive cell spreading than the other coated discs and non-coated Ti6A4V discs, which indicated that the biocompatibility of the Ti-discs can be improved using HApBGCsHep coatings. 

### 3.3. Electrochemical Test

Figure 7 shows the potentiodynamic polarization (anodic or cathodic) curves of S_1_ (control group, uncoated Ti6Al4V), S_2_ (HAp+bioglass), S_3_ (chitosan+heparin), and S_4_ (HAp+bioglass+chitosan+heparin) samples. Both anodic and cathodic polarization studies were performed at room temperature in HBSS for 1 h after the open circuit potential was taken. The potential range to be scanned was set at −200 mV to 200 mV of corrosion potential, with the scanning rate set at 1 mVs^−1^. The potentiodynamic polarization results (electrochemical parameters) of the samples are given in Table 2. According to Figure 7, the corrosion values of S_1_, S_2_, S_3_, and S_4_ samples differ from −0.457 to −0.365 V. From Figure 7 and Table 2, one can see that the E_corr_ of all the coatings is more noble than the Ti6Al4V substrate. The E_corr_ values of S_1_, S_2_, S_3_ and S_4_ samples were measured as −0.457, −0.415, −0.400 and −0.365 V, respectively. The higher the E value, the better the corrosion resistance of the materials. A significant positive shift has been found, indicating a lower corrosion tendency than that of the other specimens. This may be attributed to the thick and dense oxide coating and the appropriate content of HApBG. The results showed that the corrosion current density (icorr) of the S_1_, S_2_, S_3_ and S_4_ is − 2.51, − 0.63, 0.32 and 0.06 µA/cm^2^, respectively. The lower i_corr_ values of HApBGCsHep (S_4_) indicate lower corrosion rates [41]. This is due to the HApBGCsHep-coated sample exhibiting a good uniform film with more pits than the CsHep-coated alloy. Overall, it can be concluded from the polarization evaluations that the sample with the HApBGCsHep coating effectively improved corrosion resistance.

## 4. Conclusions

In this study, EPD and dip-coating processes were used to create multilayer coatings from HAp, BG, CS, and heparin solution. The effect of HApBG, CsHep, and HApBGCSHep multilayer coatings on the corrosion resistance, in vitro biological performance, and microstructure was investigated.

Doped HAp components remain similar in terms of chemical composition and crystallinity, regardless of the bioglass, chitosan, and heparin contents in the solution.With a proliferation rate of over 85% of the control, HApBGCSHep-coated discs proved to have greater biocompatibility than the other HAp+bioglass and chitosan+heparin coatings.The composite coatings had a smooth, crack-free surface with few micropores, according to the results of the SEM examination.Taken together, our data show that the HApBGCSHep-coated Ti6Al4V alloy is bioactive and predicts greater biocompatibility in vitro. Implantation of this composite material in animal models will be required next as proof of concept.The polarization tests revealed that the HApBGCsHep coating effectively enhanced the corrosion resistance of the coated samples.

## Figures and Tables

**Figure 1 materials-16-06608-f001:**
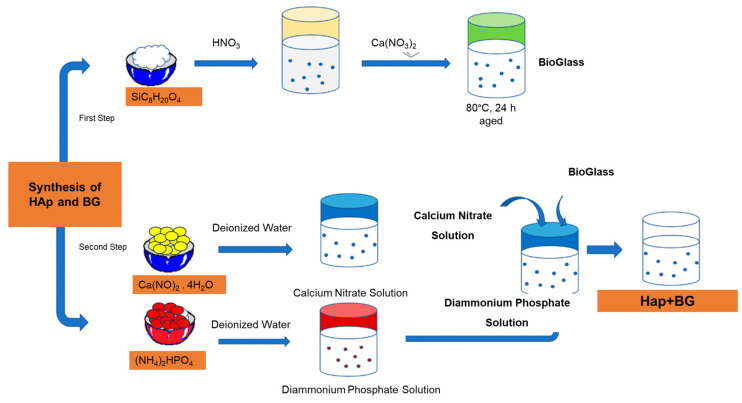
Flow chart showing the process of synthesizing of HAp and HApBG powder.

**Figure 2 materials-16-06608-f002:**
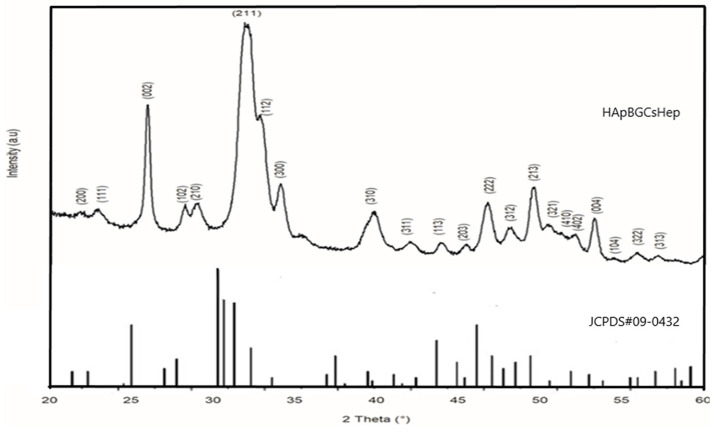
XRD pattern of S_4_ (HApBGCsHep) coating.

**Figure 3 materials-16-06608-f003:**
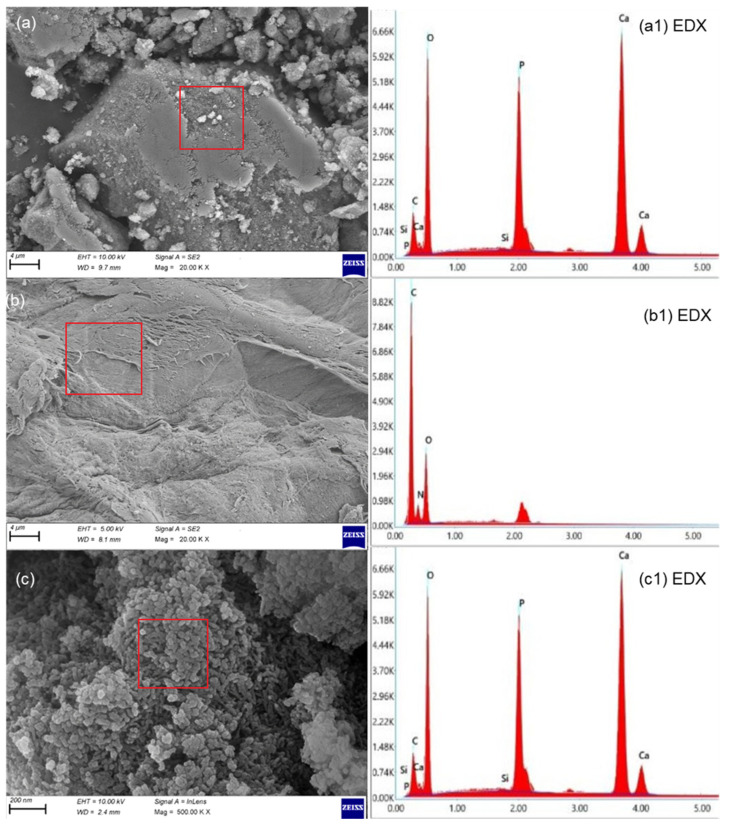
SEM and SEM-EDX spectrum of (**a**) S_2_: HA+bioglass powder, (**a1**) S_2_: HA+bioglass powder EDX (**b**) S_3_: chitosan+heparin coating, (**b1**) S_3_: chitosan+heparin coating EDX (**c**) S_4_: HA+bioglass+chitosan+heparin powder, (**c1**) S_4_: HA+bioglass+chitosan+heparin powder EDX.

**Figure 4 materials-16-06608-f004:**
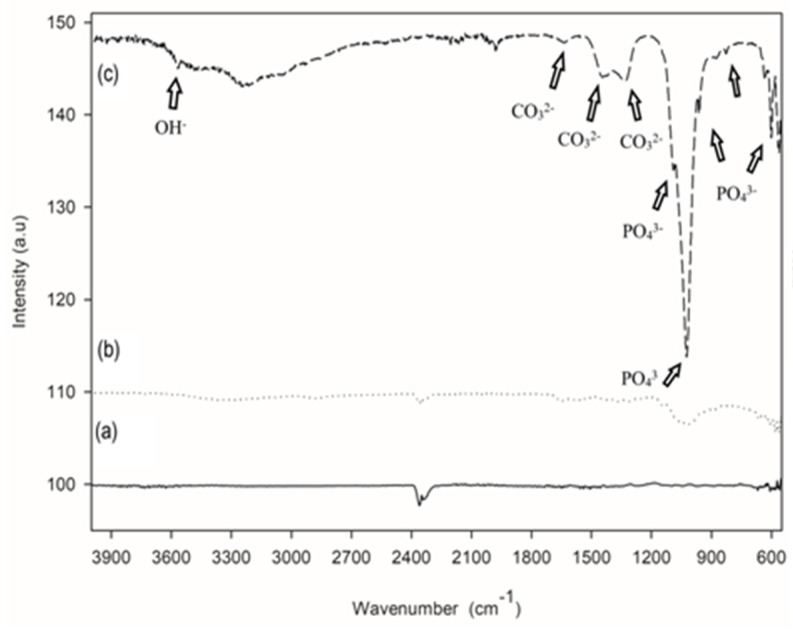
FTIR spectra of (**a**) S_2_: HA+bioglass, (**b**) S_3_: chitosan+heparin, (**c**) S_4_: HA+bioglass+chitosan+heparin coatings.

**Figure 5 materials-16-06608-f005:**
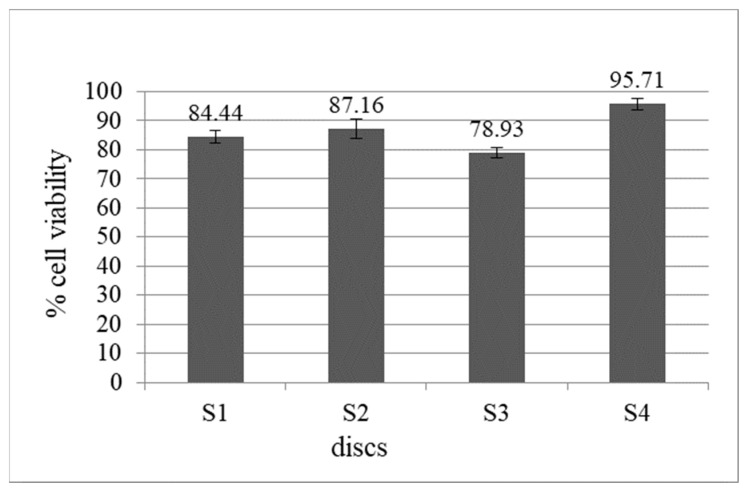
Cell viability of coated discs ((S_1_ (uncoated), S_2_ (HAp+bioglass), S_3_ (chitosan+heparin), S_4_ (HAp+bioglass+chitosan+heparin)). Values were expressed as the mean of five replicates (*n* = 5).

**Figure 6 materials-16-06608-f006:**
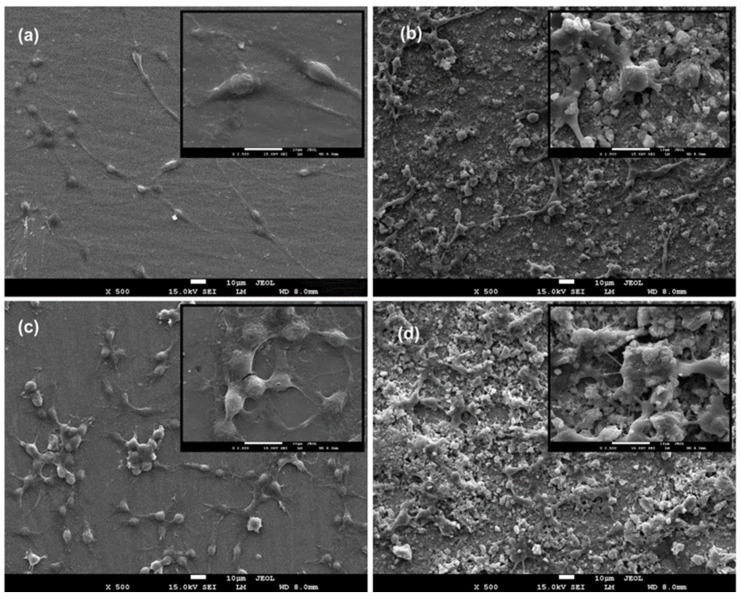
SEM morphologies of the 3T3 cells seeded on discs: (**a**) S_1_ (uncoated), (**b**) S_2_ (HAp+bioglass), (**c**) S_3_ (chitosan+heparin) and (**d**) S_4_ (HAp+bioglass+chitosan+heparin). Magnification is ×500 for large images and ×2500 form small images on top of each.

**Figure 7 materials-16-06608-f007:**
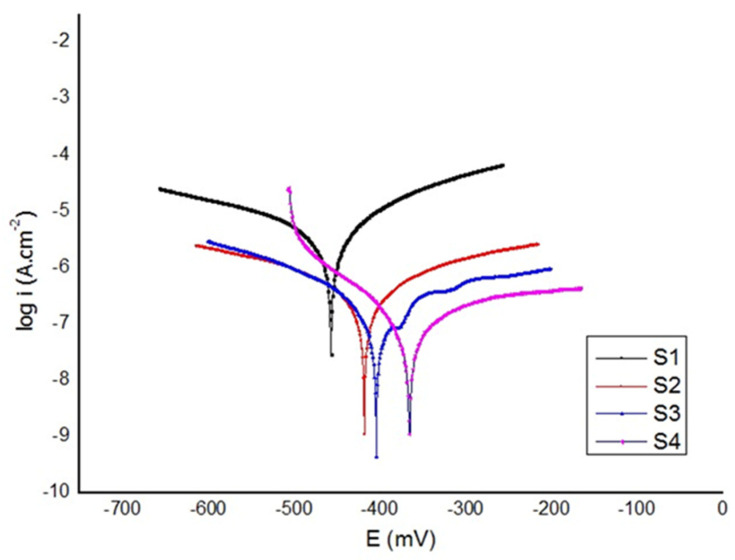
Potentiodynamic polarization curves for HAp, BG, Cs and Hep coatings. (S_1_ (uncoated), S_2_ (HAp+bioglass), S_3_ (chitosan+heparin) and S_4._ (HAp+bioglass+chitosan+heparin)).

**Table 1 materials-16-06608-t001:** Composition of Hank’s solution.

Substance	Composition (gL^−1^)
NaCl	8
KCl	0.4
NaH_2_PO_4_.2H_2_O	0.25
NaHCO_3_	0.35
Na_2_HPO_4_.2H_2_O	0.06
CaCl_2_.2H_2_O	0.19
MgCl_2_.6H_2_O	0.4
MgSO_4_.7H_2_O	0.06
Glucose	1

**Table 2 materials-16-06608-t002:** Corrosion resistance parameters obtained from potentiodynamics curves.

Material No	Material	E_corr_ (V)	i_corr_ (µA.cm^−2^)	Corrosion Rate (µm/yr)
S_1_	Bare Ti6Al4V	−0.457	2.51	24.18
S_2_	HA+Bioglass (HApBG)	−0.415	0.63	6.11
S_3_	Chitosan+Heparin (CsHep)	−0.400	0.32	3.1
S_4_	HA+Bioglass+Chitosan+Heparin (HApBGCsHep)	−0.365	0.06	0.5

## Data Availability

The data used to support the findings of this study are included within the article.

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
