# Peer review of "Fabrication and Characterization of a Multifunctional Coating to Promote the Osteogenic Properties of Orthopedic Implants"

_materials, 2023, doi:10.3390/ma16196608_

Round 1
Reviewer 1 Report
Summary
The article “Fabrication of multifunctional coating to promote osteogenic and antibacterial properties of orthopedic implants” presents the fabrication and testing of a complex coating based on hydroxyapatite, chitosan, bioglass and heparin with potential use on titanium based implants. The coating aims to improve the biocompatibility of the material by improving NIH 3T3 cells viability and reducing the risk of thrombosis and also reduce the corrosion speed of the metallic implant.
General comments
Improving metallic (titanium) implants biocomaptibility and corrosion resistance with different coatings is a subject that has been studied and developed for some time. The manuscript is well structured and shows some promising results. However, in this reviewers opinion, there is not much novelty in this work, there are some issues with the experimental work and results and the cited articles in the reference section are a bit outdated.
Specific comments
Introduction:
Please brake down large sections of text with paragraphs for a better legibility.
A large section of the introduction is dedicated to thrombosis. Although the coating includes heparin, no thrombosis testing was presented in the results.
Materials and Methods, section 2.3. Preparation of Coatings:
There are two sections of text that are repeated.
Page 3 line 132: The dimensions of the plates are given as “20x20x20 mm3” and the thickness of 0.2 cm. Please correct to represent the reality.
Page 3 line 136: You state: “EPD coating was performed in suspensions prepared by HAp, HApBG, and HApBG+CsHep.” and at line 140 “ BGHAp was initially coated on the Ti6Al4V alloy via EPD […]Finally, HApBG coated substrates were dip coated in the Cs-Hep solution and air dried” Please clarify the experimental process. Why did you use dip coating for Cs-Hep solution after you state that you used EPD?
Figure 1 is not easy to read.
Result and Discussion
Figure 2: It would be useful if you superimposed the HAp spectrum JCPDS # 09-0432 over your results.
Figure 3: It is not clear if the SEM images were performed on powders as stated on page 6 line 212 or on coatings stated on page 7 line 220. If the SEM images are on powders, spherical conglomerates can be seen in Figure 3 c. If the SEM images are on coatings, cracks are clearly visible in Figure 3 a and c. In this reviewers opinion, Figure 3 is an amalgamation of powder and coating images, Figures 3 (a, c) representing powders, and Fig 3 B representing a coating. Either way, individual elements cannot be detected on backscatter electron images.
Page 7 lines 223 - 230. This only proves the presence of apatite in the coatings. No word about the other components of the coating
Figure 4 is unclear. It would be more useful if the spectra were superimposed. Also, what is the peack at around 2400 cm-1 present in Fig 4 a and b but absent from fig 4 c.
Page 9 line 275: “According to Zheng et al. [32] the porous surface of the biomaterial is a promising surface topography” This is correct, however you have no roughness tests performed on your samples
Page 10 line 280: “Wang et al. [33] reported that cells preferentially adhered well on surfaces that exhibit a high wettability and surface energy.” This is controvential. Different cell types prefer different surfaces. Also, you do not provide wettability testing on your samples.
Section 3.2. In vitro cell behaviour (Figure 6). Why on sample S3 cell proliferation seems to be better than on sample S1, but the viability was lower?
No thrombosis testing was presented, nor the influence of different coating components was clearly presented.
Section 3.3. Electrochemical test
The electrochemical testing and results are flawed.
Why did you use the interval 350 mV below and 1000 mV above the corrosion potential? The usual interval is +/- 200 mV vs open circuit potential (OCP).
The water splitting potential is 1.23 V and the process surely interferes with your results.
The zigzag line (noise or corrosion processes) in the cathodic reaction zone could interfere with the fitting process of the data.
The icorr data in Table 2 could not be calculated from the data shown in Figure 7. Also, in Fig. 7 current density can never be negative. There is no SI unit named “mil”, you probably meant mm.
Based on the experiments, the conclusions are correct, however more emphasis should be made on cell viability and potential uses of the coating.
English language is fine. Some minor spelling errors were detected in the text.
Author Response
Specific comments
Introduction:
- Please break down large sections of text with paragraphs for a better legibility.
- Response: As reviewer stated large sections divided into paragraphs.
- A large section of the introduction is dedicated to thrombosis. Although the coating includes heparin, no thrombosis testing was presented in the results.
- Response: The authors appreciate the valuable comment of the reviewer. As heparin is well-known for its anticoagulant activity [Zare et al., 2024], the present study was conducted on the biocompatibility of the modified-Ti6Al4V alloy. Comprehensive biological activity studies will be done for further studies, especially as in vivo experiments.
- [Zare, E. N., Khorsandi, D., Zarepour, A., Yilmaz, H., Agarwal, T., Hooshmand, S., ... & Khademhosseini, A. (2024). Biomedical applications of engineered heparin-based materials. Bioactive Materials, 31, 87-118.]
Materials and Methods,
Section 2.3. Preparation of Coatings:
- There are two sections of text that are repeated.
- Response: Repetitive text has been removed.
- Page 3 line 132: The dimensions of the plates are given as “20x20x20 mm3” and the thickness of 0.2 cm. Please correct to represent the reality.
- Response: The dimensions of 20x20x20 mm3 modified as with cross-section of 20x20 mm2
- Page 3 line 136: You state: “EPD coating was performed in suspensions prepared by HAp, HApBG, and HApBG+CsHep.” and at line 140 “ BGHAp was initially coated on the Ti6Al4V alloy via EPD […]Finally, HApBG coated substrates were dip coated in the Cs-Hep solution and air dried” Please clarify the experimental process. Why did you use dip coating for Cs-Hep solution after you state that you used EPD?
- Response: Line 133-134 sentence ‘EPD coating was performed in suspensions prepared by HAp, HApBG, and HApBG+CsHep’ changed as ‘EPD coating was performed in suspensions prepared by HAp, HApBG (Figure 1) and HApBG+CsHep (CsHep dip coated)’.
- Figure 1 is not easy to read.
- Response: Figure 1 changed as flow chart showing the process of synthesizing of HAp and HApBG powder
Result and Discussion
- Figure 2: It would be useful if you superimposed the HAp spectrum JCPDS # 09-0432 over your results.
- Response: Figure 2, HAp spectrum JCPDS # 09-0432 superimposed over our result.
- Figure 3: It is not clear if the SEM images were performed on powders as stated on page 6 line 212 or on coatings stated on page 7 line 220. If the SEM images are on powders, spherical conglomerates can be seen in Figure 3 c. If the SEM images are on coatings, cracks are clearly visible in Figure 3 a and c. In this reviewer’s opinion, Figure 3 is an amalgamation of powder and coating images, Figures 3 (a, c) representing powders, and Fig 3 B representing a coating. Either way, individual elements cannot be detected on backscatter electron images.
- Response: The authors appreciate the valuable comment of the reviewer. We changed the description of the figure 3. As figure 3a and figure 3c performed on powder and figure 3b performed on coated surface.
- Page 7 lines 223 - 230. This only proves the presence of apatite in the coatings. No word about the other components of the coating.
- Response: The authors appreciate the valuable comment of the reviewer but we couldn’t understand this part exactly.
- Figure 4 is unclear. It would be more useful if the spectra were superimposed. Also, what is the peak at around 2400 cm-1present in Fig 4 a and b but absent from fig 4 c.
- Response: FTIR spectra were superimposed in figure 4. The transmission of the HABGCsHep four times bigger than HABG. Since HABG not obvious in graph, we gave separate graphs. We scanned the literature but we couldn’t define the peak at 2400 cm-1
- Page 9 line 275: “According to Zheng et al. [32] the porous surface of the biomaterial is a promising surface topography” This is correct; however, you have no roughness tests performed on your samples.
- Response: The reference has been removed from the text. The authors appreciate the valuable comment of the reviewer. Comprehensive characterization and biological activity studies will be done for further studies, especially as in vivo experiments.
- Page 10 line 280: “Wang et al. [33] reported that cells preferentially adhered well on surfaces that exhibit a high wettability and surface energy.” This is controvential. Different cell types prefer different surfaces. Also, you do not provide wettability testing on your samples.
- Response: The reference has been removed from the text. The authors appreciate the valuable comment of the reviewer. Comprehensive characterization and biological activity studies will be done for further studies, especially as in vivo experiments.
- Section 3.2. In vitro cell behaviour (Figure 6). Why on sample S3 cell proliferation seems to be better than on sample S1, but the viability was lower?
- Response: Required discussion has been added to the manuscript.
- No thrombosis testing was presented, nor the influence of different coating components was clearly presented.
- Response: The authors appreciate the valuable comment of the reviewer. Comprehensive characterization and biological activity studies will be done for further studies, especially as in vivo experiments.
Section 3.3. Electrochemical test
- The electrochemical testing and results are flawed.
- Response: The authors appreciate the valuable comment of the reviewer. All electrochemical measurements were reviewed and TAFEL measurements of samples S3 and S4 were taken again.
- Why did you use the interval 350 mV below and 1000 mV above the corrosion potential? The usual interval is +/- 200 mV vs open circuit potential (OCP).
- Response: We wanted to examine the anodic part to see the anodic dissolution of the samples. As you stated, the standard measurement should be +/-200 mV against the open circuit potential. Since the results did not differ much in the uncoated titanium alloy, we used the same measurement but performed the others according to the standard measurement procedure.
- The water splitting potential is 1.23 V and the process surely interferes with your results.
- Response: The authors appreciate the valuable comment of the reviewer. As reviewer stated we corrected all results.
- The zigzag line (noise or corrosion processes) in the cathodic reaction zone could interfere with the fitting process of the data.
-Response: The authors appreciate the valuable comment of the reviewer. Taking this information into consideration, we repeated our measurements
- The icorr data in Table 2 could not be calculated from the data shown in Figure 7. Also, in Fig. 7 current density can never be negative. There is no SI unit named “mil”, you probably meant mm.
- Response: We rearranged Table 2 according to the data obtained from figure 7. In Fig. 7’ de the “current density “was changed as “log current density”.
- Based on the experiments, the conclusions are correct, however more emphasis should be made on cell viability and potential uses of the coating.
Response: Conclusion has been revised according to the Reviewer’s comment
PS: The english language was checked by a native speaker.

Reviewer 2 Report
In their manuscript entitled “Fabrication of multifunctional coating to promote osteogenic and antibacterial properties of orthopedic implants”, the authors describe the preparation of composite coatings of ceramics (hydroxyapatite/bioglass) and polysaccharides (chitosan/heparin) on the titanium alloy Ti6Al4V. The aim of the study is a bilayer coating with the aim to enhance the biocompatibility of cp Ti. The coatings are analyzed by SEM/EDX and ATR-FTIR but miss the complementary information on roughness, wettability and charge, which are known to contribute to the behavior of implants in their natural environment. Despite of the lack of such comprehensive characterization, the manuscript is well structured and the physical characterization of the coatings is complemented by the biological assays of cell compatibility and proliferation.
I have the following comments for consideration in a minor revision of the manucript.
The antibacterial properties are not investigated in this paper. Although chitosan with its known antimicrobial behavior is incorporated in the multifunctional bilayer coating, it is not self-evident that it does exhibit its properties in the heparin matrix. I therefore suggest to either exclude the term “antibacterial” from the title or to add the respective experimental assessment.
The keywords refer to the ingredients of the multifunctional coating. I suggest to add “Ti6Al4V alloy” as an another keyword to emphasize the application of these coatings to titanium-based biomaterials.
Lines 131-139 are a duplicate of lines 123-131.
A modified display of the chemical formulas in lines 107 and 108 is recommended to reflect the compositions of these chemicals.
The dimensions of the Ti plates in line 124 refer to a cube while the thickness of 0.2 cm suggests a cuboid (platelet). I assume that the cross-section of Ti is 20 x 20 mm2.
The display of the pictures at the bottom of Figure 1c is too small.
The authors are encouraged to discuss the possible reason for the lower cell viability fo S3 in comparison with S4 (here the CsHep coatings are supposed to stay on the top surface).
The authors should mention the origin of the error bars in Figure 5. Are these calculated from duplicate or triplicate measurements?
The abscissa in Figure 7 should display mV instead of V. Furthermore, it is difficult to identifiy the peak values in current density. A magnification of the region -750…0 mV is helpful.
I suggest the authors to revise the English grammar of their manuscript. Some articles or verbs are occasionally missing, e.g., lines 20 and 24.
Author Response
We would like to thank both reviewers for taking the time to thoroughly read our study and for their constructive, positive feedback, which substantially improves the quality of the paper. In the following, detailed responses to the reviewers’s comments/corrections can be found.
- The antibacterial properties are not investigated in this paper. Although chitosan with its known antimicrobial behavior is incorporated in the multifunctional bilayer coating, it is not self-evident that it does exhibit its properties in the heparin matrix. I therefore suggest to either exclude the term “antibacterial” from the title or to add the respective experimental assessment.
- Response: The term “antibacterial” has been excluded from the title.
- The keywords refer to the ingredients of the multifunctional coating. I suggest to add “Ti6Al4V alloy” as another keyword to emphasize the application of these coatings to titanium-based biomaterials.
- Response: Keywords have been re-written.
- Lines 131-139 are a duplicate of lines 123-131.
- Response: Repetitive text has been removed.
- A modified display of the chemical formulas in lines 107 and 108 is recommended to reflect the compositions of these chemicals.
- Response: Formulas in lines 110-113 modified as:
In order to produce the doped HA coating, the following chemicals were used: Ca(NO3)2.4H2O (Isolab, ≥ 99%), SiC8H20O4 (Merck, ≥ 99), HNO3 (Sigma-Aldrich, ≥ 65%), C6H15O4P (Alfa Aesar, ≥ 98%), C2H6O (Sigma-Aldrich, ≥ 99.8%), (C6H11NO4)n chitosan (TCI), heparin (Kocak Farma), and double-distilled water.
- The dimensions of the Ti plates in line 124 refer to a cube while the thickness of 0.2 cm suggests a cuboid (platelet). I assume that the cross-section of Ti is 20 x 20 mm2.
- Response: The dimensions of 20x20x20 mm3 modified as with cross-section of 20x20 mm2.
- The display of the pictures at the bottom of Figure 1c is too small.
- Response: Figure 1 changed as flow chart showing the process of synthesizing of HAp and HApBG powder.
- The authors are encouraged to discuss the possible reason for the lower cell viability for S3 in comparison with S4 (here the CsHep coatings are supposed to stay on the top surface).
- Response: Required discussion has been added to the manuscript.
- The authors should mention the origin of the error bars in Figure 5. Are these calculated from duplicate or triplicate measurements?
- Response: Cell viability has been calculated over five samples (n=5). Required information has been added to the figure caption
- The abscissa in Figure 7 should display mV instead of V. Furthermore, it is difficult to identifiy the peak values in current density. A magnification of the region -750…0 mV is helpful.
- Response: Figure 7 was rearranged and X axis changed to mV.
PS:The English language was checked by a native speaker.

Reviewer 3 Report
Please find the attached file.

Quality fo English is satisfactory.
Author Response
We would like to thank both reviewers for taking the time to thoroughly read our study and for their constructive, positive feedback, which substantially improves the quality of the paper. In the following, detailed responses to the reviewers’s comments/corrections can be found.
- The text of figure 1 is not clear. Rewrite/redraw the figure.
- Response: Figure 1 changed as flow chart showing the process of synthesizing of HAp and HApBG powder.
- To improve the surface bioactivity and reduce the corrosion rate of the Ti6Al4V alloy, why multi-layered coatings (HAp, BG, Cs, Hep) were required.
Multi-layered coatings were required for the better integration of the implant and the bone. As stated in the manuscript line 68-73:
‘Despite their wide range of applications, metallic implant defects and failure arise due to inadequate mechanical bonding, postoperative clotting problems, aseptic loosening and infections [14]. The main reason of aseptic loosening is implant micromotion owing to gaps between implant-tissue interface. Therefore, the application of ceramic/biopolymer surface coating is intended to increase the osteointegration properties of metallic implants. Bioactive materials like Hap and chitosan form bioactive bonding with bone and implant.
- How they impact on the surface bioactivity.
- Sometimes, an intermediate TiO2layer is produced between the Ti and HA layer for improving the bonding capacity of HA on the Ti substrate and, thus, improving the integrity of the coating. Also hydroxyapatite coating decreases the release of metallic ions by acting like a barrier between hard tissue and implant material.
- As stated in the lines 81-84:
‘Bioglasses produces specific biological responses at in-vivo interfaces between soft tissue and bone. Especially, bioactive glasses containing CaO-SiO2-P2O5 can be attached to soft and hard tissues without intermediate fibrous layers’
- In addition, how much thickness was used of each layer using electrophoretic deposition.
In this study, EPD coating of HA+BG by voltage 30V for 1minutes the coating thickness of HA+BG was about 50 µm. With dip-coating coating thickness changed (>50 µm).
- Why EPD method is used? How EPD is preferable to other deposition methods?
As stated in the manuscript lines 95-98:
‘In recent years, electrophoretic deposition (EPD) used to produce uniform coating at low temperature. The significance of this technique comes from its simplicity, low equipment cost, easy control of thickness, possibility to be applied on complex shaped implants and porous structures [19]’
- Which XRD, SEM and EDX were used for characterization? Write the model number and company name. In addition, provide all the company data of the materials used in the study.
For XRD, SEM and SEM-EDX model number and company name added:
- XRD (PANalytical Empyrean, The Analytical X-ray Company)).)
- SEM (SEM- Zeiss, Gemini Sigma 300)
- SEM- EDX (SEM- Zeiss, Gemini Sigma 300)
- EDX data for figure 3 is blur, change the figure with clear and readable numbers and texts.
- Response: EDX figure has been checked and changed.
- Discuss in detail the co-efficient of performance (%CV) for figures 5.
- Response: Required discussion has been added to the manuscript.
- Figure 5 also needs more discussion.
- Response: Required discussion has been added to the manuscript.
- For figure 6, the magnification and other data is not readable. Change it with a clear scale bar.
- Response: Images has been checked and changed.
- Potentiodynamic polarization curves for HAp,BG,Cs and Hep coatings. (S1(uncoated), S2 (HAp+Bioglass), S3 (Chitosan+Heparin) and S4(HAp+Bioglass+Chitosan+Heparin)) needs more scientific discussion.
- Response: Required discussion has been added to the manuscript.
- I encourage the authors that they could give the following recent studies “Recent Developments in Zn-Based Biodegradable Materials for Biomedical Applications; Performance analysis of biodegradable materials for orthopedic applications; Recent developments in coatings for orthopedic metallic implants” their introduction part.
- Response: Required references have been added to manuscript.
PS: The English language was checked by a native speaker.

Round 2
Reviewer 1 Report
Dear Authors,
Thank you for responding to the raised questions. This reviewer is satisfied with the current form of the manuscript. Good luck with your further research!
Reviewer 3 Report
Paper is significantly improved.